# Excess mortality during the COVID-19 pandemic (2020–2021) in an urban community of Bangladesh

**Mohammad Sorowar Hossain**[1,2☯*], **Jahidur Rahman Khan**[1☯], **S. M. Abdullah Al Mamun**[1], **Mohammad Tariqul Islam**[3], **Enayetur Raheem**[1]

1 Department of Emerging and Infectious Diseases, Biomedical Research Foundation, Dhaka, Bangladesh,
2 School Life Environment and Life Sciences, Independent University, Bangladesh, Dhaka, Bangladesh,
3 Jamalpur General Hospital, Jamalpur, Bangladesh

☯ These authors contributed equally to this work.
* sorowar.hossain@brfbd.org

## Abstract

Measuring COVID-19-related mortality is vital for making public health policy decisions. The magnitude of COVID-19-related mortality is largely unknown in low- and middle-income countries (LMICs), including Bangladesh, due to inadequate COVID-19 testing capacity and a lack of robust civil registration and vital statistics systems. Even with the lack of data, cemetery-based death records in LMICs may provide insightful information on potential COVID-19-related mortality rates; nevertheless, there is a dearth of research employing cemetery-based death records. This study aimed to assess the excess mortality during the COVID-19 pandemic in an urban setting in Bangladesh using a cemetery-based death registration dataset. A total of 6,271 deaths recorded between January 2015 and December 2021 were analysed using a Bayesian structural time series model. Exploratory analysis found that the average monthly number of deaths was 69 during the pre-COVID-19 period (January 2015-February 2020), but significantly increased to 92 during the COVID-19 period (March 2020-December 2021). The increase in male deaths was twice as large as the increase in female deaths. Model-based results were not statistically significant (relative effect 17%, 95% credible interval: -18%, 57%), but there was an overall increasing trend during the COVID-19 period, and specific months or shorter periods had a substantial increase. This first-of-its-kind study in Bangladesh has assessed the excess mortality in an urban community during the COVID-19 pandemic. Cemetery-based death registration appears to aid in tracking population mortality, especially in resource-limited countries where collecting data on the ground is challenging during crisis periods; however, additional large-scale research is required.

## Background

The rapid spread of COVID-19 caused a considerable number of deaths worldwide. However, a clear picture of the magnitude of fatalities is lacking and varies between low- and middle-

**Data Availability Statement:** All data used are available in the manuscript.

**Funding:** The authors received no specific funding for this work.

**Competing interests:** The authors have declared that no competing interests exist.

income countries (LMICs) and high-income countries (HICs). This may be attributable to differences in population dynamics, the implementation of COVID-19-related restrictions, health system capacity, and comprehensive civil registration and vital statistics systems. Understanding the magnitude of COVID-19-attributed mortality is crucial for public health policy decisions to reduce mortality and prevent future crises. Nearly 80% of the world's population lives in LMICs [1]. Unlike HICs, underreporting deaths is a genuine concern for most LMICs due to inadequate COVID-19 testing capacity and a lack of robust civil registration and vital statistics systems (CRVS) [2]. Consequently, the extent of COVID-19-related mortality is unclear in many LMICs, including Bangladesh.

Bangladesh, with a population of over 165 million, is one of the most densely populated countries in South Asia [3]. The Department of Economic and Social Affairs of the United Nations estimates that 3 million births and 0.9 million deaths occur annually in this country [4]. The coverage of deaths in the existing CRVS is very low (only 17%) [5]. Even though hospitals are the primary source of cause-specific death information, they are not integrated into the CRVS. Due to the lack of a proper medical record-keeping system, the validity of the hospital-issued death certificate remains questionable [5, 6]. Importantly, most deaths occur at home in Bangladesh, and therefore the cause of death is primarily unknown [5]. Since the year 2020, Bangladesh has experienced repeated waves of COVID-19. The government implemented movement restrictions, which led to significant limits on the accessibility of healthcare services [7, 8]. These may result in an increase in all-cause mortality during the COVID-19 pandemic. Assessing excess mortality is a useful technique for gaining a better understanding of the potential deaths associated with COVID-19. However, it is important to note that not all deaths are a direct result of COVID-19, but rather indirect effects or other causes. For example, COVID-19-related restrictions and strained healthcare systems may increase noncommunicable disease (NCD)-related excess mortality. Excess mortality is defined as the observed death from any cause relative to the predicted deaths based on historical averages.

In the case of Bangladesh, two community-based studies have thus far estimated the death toll in rural communities during the early months of the COVID-19 pandemic in 2020 [9, 10]. In the first wave of the pandemic, the infection rate and decoded official deaths were higher among urban areas compared to rural areas [9, 10]. Bangladesh lacks information on the burden of the delta variant with a surge in cases and deaths in 2021. Moreover, previous studies were limited to rural communities, relied on surveillance and survey data, evaluated an earlier period of the pandemic, and did not apply time-series analysis based on a counterfactual scenario, therefore lacking a comprehensive scenario of excess mortality during the COVID-19 pandemic.

In Bangladesh, a predominantly Muslim nation, burials are typically conducted in cemeteries, the locations of which are determined by factors including proximity, available space, and family preference. Therefore, cemetery-based death registration datasets are an important source for providing information about excess deaths concurrently with other death registration systems.

There has been paucity of studies that use cemetery-based data to understand the excess mortality during COVID-19. Two cemetery-based studies in Somalia and Yemen used satellite photos to count graves and burial activities, as well as to assess excess mortality during the pandemic [11, 12]. This study aimed to assess the magnitude of excess mortality during the COVID-19 pandemic in an urban setting using a unique cemetery-based death registration dataset. Findings could be useful for the ongoing pandemic and for planning for future pandemics.

## Methods

### Study setting

Jamalpur town (also known as Pauroshova) is one of the 64 district towns in Bangladesh. It is located 142 kilometres northwest of the capital city, Dhaka. Jamalpur town's 53.3 km$^2$ area is home to more than 180,358 people (population density of 3,385/km$^2$) with an annual growth rate of 1.7% based on available 2015 census data [13]. This medium-sized urban setting is socio-demographically similar to other 33 district towns in Bangladesh [14].

### Cemetary-based registry dataset

British India's colonial administration created a birth, death, and marriage registration system. Until 2004, Bangladesh used the registration system (for birth, death, and marriage) created by British India's colonial administration in the late nineteenth century [5]. The new CRVS system became functional in 2010 after the introduction of the online-based birth registration system. However, death can't be registered through an online system; it can be done directly by visiting a registration office (such as Pauroshova). While a death certificate is required for property succession, claiming a pension and insurance, and making a specified bank deposit, families from lower socioeconomic backgrounds are less likely to register a death [5]. Most importantly, because of space limitations and expensive land prices in district towns, practically all families opt to bury their loved ones in public cemeteries. People in rural regions, on the other hand, often bury the deceased in private cemeteries. Even if individuals relocate outside the community (such as in cities) for work, they are usually buried in the local community cemetery when they die.

Jamalpur town authority collects self-reported data on death cases registered at the existing four Muslim cemeteries (around 95% of the people in this community are Muslim) and one Hindu cremation site [13]. We extracted available data (age, sex, and month of death) from the Pauroshova death registry books for individual death cases. The dataset contained 10,482 registered deaths between January 2011 and December 2021. A total of six records were removed due to incomplete data, resulting in 10,476 deaths in the dataset. This study examined 7,300 deaths (4,409 male and 2,891 female) across all age groups that occurred between January 2015 and December 2021. The main analysis of this study focused on 6,271 deaths of individuals aged 35 years or older (3,790 male and 2,481 female) recorded between January 2015 and December 2021 (Fig 1). This study restricted age groups over 35 years because over 96% of COVID-19 related official deaths occurred at that age [15]. However, a sensitivity analysis was performed, which included deaths from all age groups. The outcome measure, i.e., the monthly count of the total number of deaths from any cause, was computed by summing the individual records for each month. The exposure period (i.e., COVID-19) was defined as March 2020 to December 2021, and the pre-exposure period (i.e., the pre-COVID-19 period) was defined as January 2015 to February 2020. Notably, the COVID-19 period was set to start in the same month that the WHO declared COVID-19 a pandemic (March 2020). This resulted in 62 pre-COVID-19 months and 22 COVID-19 months in total.This study was approved by the Institutional Review Board at Biomedical Research Foundation, Bangladesh (BRF/ERB/2020/E03).

### Statistical analysis

We calculated the monthly average number of deaths between the pre-COVID-19 and COVID-19 eras. In addition, we calculated the difference in monthly average deaths between these two periods and carried out the Wilcoxon rank-sum test to determine the statistical significance of this difference. The potential impact of COVID-19 on mortality was further

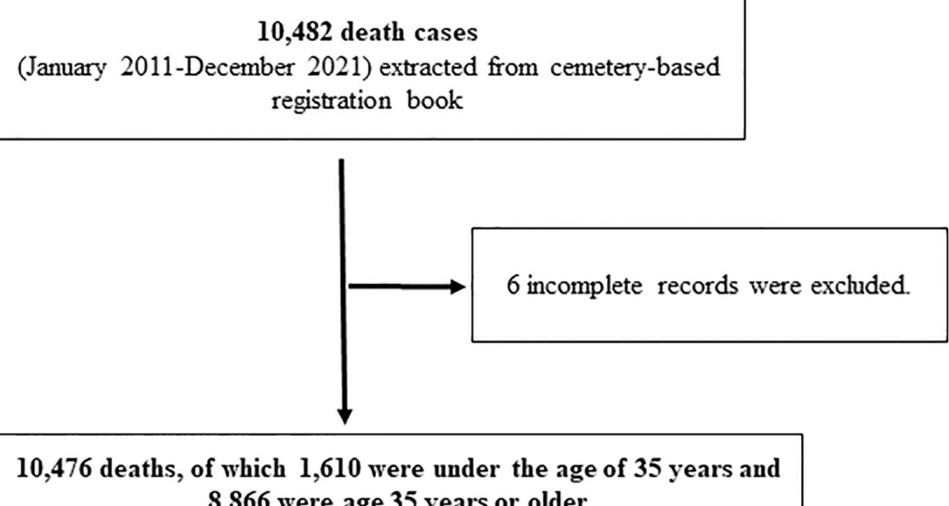

**Fig 1. Schematic description of the cemetery-based dataset.**

evaluated in this research by using a Bayesian structural time series (BSTS) model, which has been described in detail elsewhere [16–18] In brief, it is a stochastic state-space model that can incorporate trend, seasonality, and regression components [16–18]. A pair of equations can be used to describe this model,

$$\text{Observation equation}: y_t = Z_t^T \alpha_t + \epsilon_t$$

$$\text{Transition equation}: \alpha_{t+1} = T_t \alpha_t + R_t v_t$$

This observation equation establishes a connection between observed data ($y_t$, e.g., number of deaths) and a latent state ($\alpha_t$); and the transition equation explains how latent states ($\alpha_t$) evolve over time. $Z_t$ represents the output vector, $T_t$ the transition matrix, $R_t$ the control matrix, $\epsilon_t$ the observation error, and $v_t$ system error.

The impact of intervention (e.g., COVID-19 period) on the outcome (e.g., mortality) can be inferred based on the findings of this model. In a variety of public health settings, this model has already been used to measure the impact of various interventions, including COVID-19

pandemic. For example, it has been used to assess the impact of social protest on emergency health service utilization, the effect of local alcohol licencing laws on hospital admission and crime, and the effect of the COVID-19 pandemic on the decline in tuberculosis cases [19–21]. This model is quasi-experimental in nature in that it measures the impact of an intervention by predicting a counterfactual time-series (i.e., the outcome that would have occurred if no intervention had taken place). This counterfactual is predicted by assessing the behaviour of the time series. Finally, the difference between observed time-series during the intervention period to the counterfactual scenario generated by the model is used to assess the impact.

In this study, a univariate time series was considered i.e., monthly number of deaths due to a lack of covariates, meaning that the model was estimated only using the behaviour of this time-series.

The first step included estimating the model using data on the monthly number of deaths that occurred during the pre-COVID-19 period, and the second step involved using the estimated model to predict the number of deaths that would occur during the COVID-19 period if the pandemic did not occur (counterfactual). Finally, during the COVID-19 period, the difference between the predicted and actual number of deaths was evaluated to measure the effect of the pandemic. The point effects and their 95% credible intervals (CrIs) were generated by comparing predicted and actual death trends across 40,000 Markov Chain Monte Carlo iterations (10% burn-in period). This model was chosen because of its appealing properties, such as its ability to properly portray stochastic behaviour by enabling model parameters to change over time and the benefit of setting prior beliefs on the parameters (as it uses a Bayesian framework). All analyses were performed using R.

## Results

Table 1 compares the monthly average number of fatalities before and during the COVID-19 pandemic, overall and separately for males and females. During the pre-COVID-19 period, the average monthly number of deaths was 69, whereas, during the COVID-19 period, this figure significantly increased by 23 and reached 92. During the COVID-19 period, the average monthly number of male and female deaths increased significantly by 16 (from 40.9% to 57.0%) and 7 (from 27.6% to 35.0%), respectively. This indicates that males had a higher average monthly death rate than females, and that the change in male deaths during the pandemic was also greater than the change in female deaths.

Fig 2 shows the monthly number of observed and model-predicted deaths (dotted line), where Fig 3 shows the differences between observed and model-predicted deaths (dotted line). There was a notable rise in month deaths during the COVID-19 period, specific months (e.g., August-September 2020 and July 2021). The cumulative difference between observed and

**Table 1. The difference in the monthly average number of deaths between pre-COVID-19 and COVID periods, as well as a model-based estimate of the impact of COVID-19.**

| Number of deaths | Pre-COVID (Jan 2015 to Feb 2020) | During COVID (Mar 2020 to Dec 2021) | Difference[1] | Relative effect 95% Credible Interval[2] |
|---|---|---|---|---|
| Total | 68.5 | 91.9 | 23.4* | 17% (-18%, 57%) |
| Male | 40.9 | 57.0 | 16.1* | 29% (-15%, 75%) |
| Female | 27.6 | 35.0 | 7.4* | 2.9% (-61%, 70%) |

[1]Difference between the monthly average deaths,

[2]BSTS model based results on the impact of COVID-19

*p<0.001, based on Wilcoxon rank-sum test

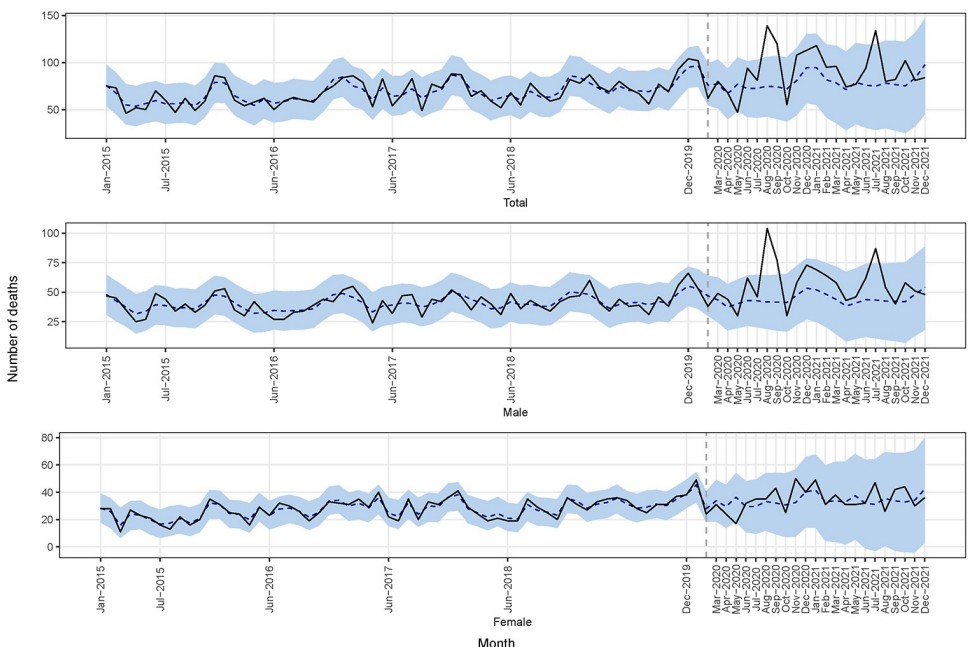

**Fig 2. The monthly number of observed and predicted deaths (total, male, and female) observed (solid line) and predicted (dotted line) time series and 95% credible intervals (blue areas) according to the Bayesian structural time series.**

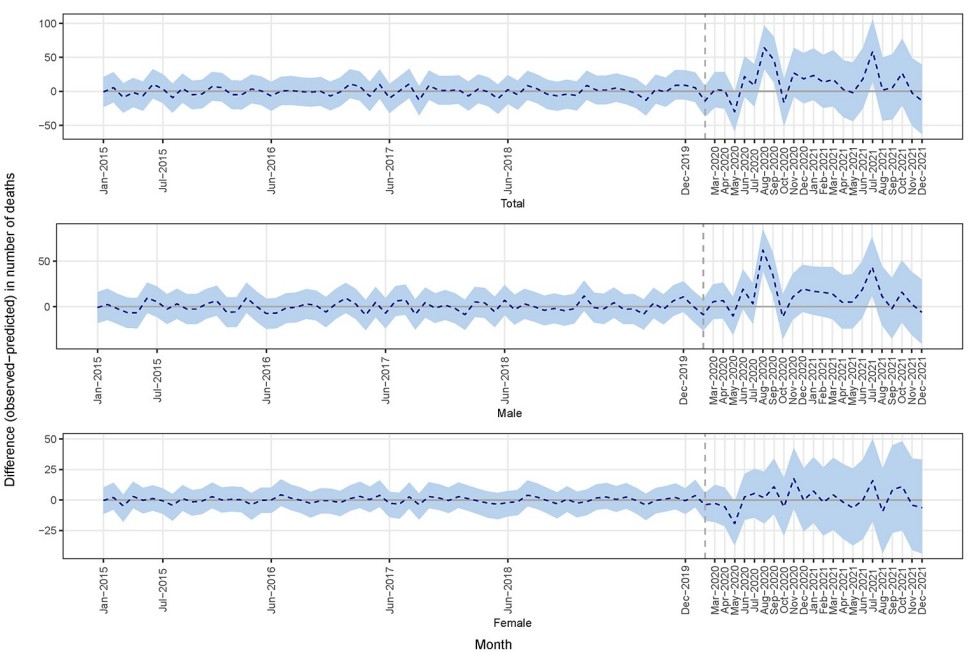

**Fig 3. Differences between observed and predicted deaths (dotted line).**

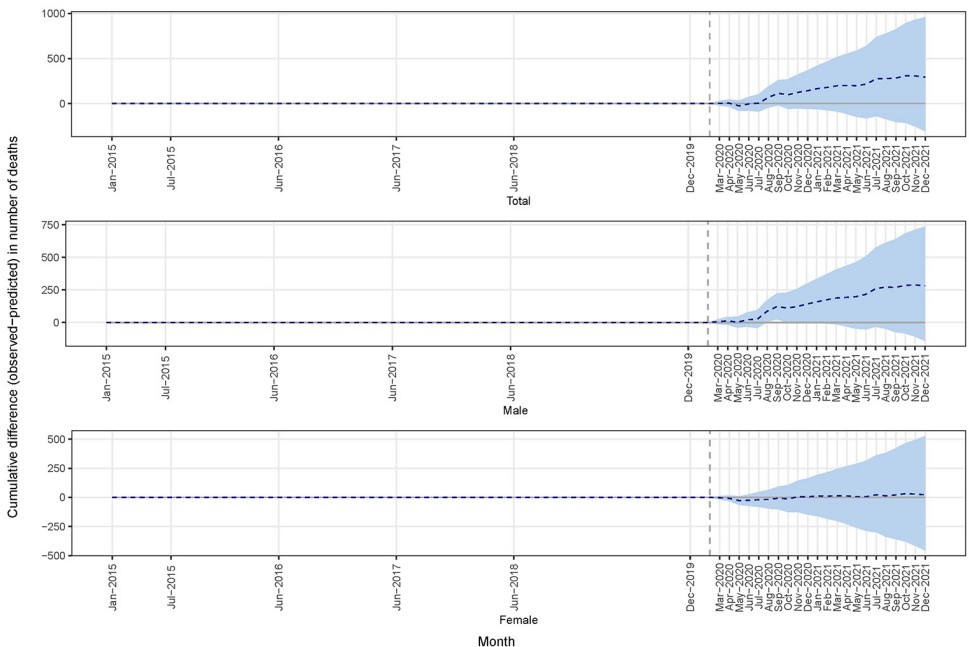

**Fig 4. The cumulative difference between observed and predicted deaths (dotted line) in the COVID period.**

predicted deaths in the COVID-19 period shown in Fig 4 indicated an increase in the number of deaths, particularly among males. The summary model-based results for the full and stratified by gender analyses are shown in Table 1. Overall, according to model-based results, during COVID-19 period, the overall number of deaths increased on average by 17% (95% CrI: -18%, 57%), 29% for males (95% CrI: -15%, 75%), and 2.9% for females (95% CrI: -61%, 70%). However, these results were not statistically significant, but the considerable increase in deaths during the COVID-19 period may be indicative of an excess of mortality. The sensitivity analysis, which encompassed deaths data from all age cohorts, yielded analogous findings (S1 Table).

## Discussion

This is, as far as we are aware, the first study in Bangladesh to examine excess mortality during the COVID-19 pandemic using cemetery-based death records in a medium-sized urban community. Our exploratory analysis revealed a significant increase in monthly mortality during the COVID-19 period (March 2020- December 2021) compared to the historical pattern of the pre-COVID-19 period (January 2015-February 2020), with the change in male fatalities being two times greater than the change in female deaths. Novel model-based results on excess mortality were not statistically different from the null for the entire COVID-19 period, but individual months or shorter periods had a significant impact. Random fluctuations of the observed series, a lengthy COVID-19 period, and the absence of sufficient control variables may have contributed to the overall significance of the findings.

In Bangladesh, two prior studies in rural settings, one based on household surveys, found a decrease in all-cause mortality in 2020 (compared to 2019), while another study based on surveillance data found higher mortality during pandemic period (i.e., 2020), particularly among 65 years or older [9, 10]. In contrast, month-specific higher excess mortality was recorded in August and September 2020 in this present study. Surprisingly, no significant excess deaths

were observed during the deadly delta variant wave in 2021 in our study setting when there was a surge of new cases and deaths across the country [7, 22]. Interestingly, in neighbouring India, COVID-19-related mortality was quite low in 2020. but with the emergence of the deadly delta variant, both cases and deaths skyrocketed in 2021. According to a recent study, states with large and mostly rural populations experienced higher excess mortality, which was up to 197% in 2021 [23].

It is important to mention that, unlike HICs, tertiary healthcare facilities necessary for managing severe COVID-19 patients are large city-based, particularly Dhaka, the capital city of Bangladesh [7, 22]. Peripheral districts of the country including our study area are not also equipped with healthcare facilities for treating other severe diseases (such as cancer, and cardiovascular diseases). Notably, over 70% of patients seek healthcare services in private hospitals and clinics in Bangladesh [24, 25]. During the early months of the COVID-19 pandemic, the country's overall healthcare system collapsed; all private hospitals and clinics were closed down because of panic and inadequate supply of PPEs [7, 26, 27]. Medical doctors faced stigmatization and people were scared of COVID-19 testing. Consequently, a substantial number of non-COVID-19-associated deaths might contribute to excess mortality in 2020. On the other hand, deaths from road accidents and occupational injuries were expected to be low because of nationwide movement restrictions and work-from-home policy. Road accidents and the resulting deaths have become a significant public health concern in recent years in Bangladesh. Despite some movement restrictions during the study period, there were still a high number of deaths reported. At least 6,686 and 7,809 road accident-related deaths were documented in 2020 and 2021 respectively [28]. However, it is important to note that the actual number of road accidents and related deaths may be even higher due to underreporting, which is a common problem in LMICs including Bangladesh. It is difficult to determine the exact number of accidents and deaths because not all accidents are reported to the authorities, and even when they are, not all of them are properly documented [29]. Since official COVID-19 cases and fatalities were largely documented in major two cities (Dhaka and Chittagram) in the early phases of the pandemic, this could the possible reason for no excess mortality reported in rural settings [7]. In 2021, most private healthcare facilities were functional which provided support for COVID-19 and other non-COVID-19 patients in the country. Moreover, Vaccination against COVID-19 is supposed to reduce the mortality. However, vaccination coverage was very low in our study period (only 4% two doses as June 2021) [30]. As a consequence, this might have impacted the overall excess death in 2021 in our study setting.

According to official documentation, male fatalities were much higher as compared to females in Bangladesh. Our finding is consistent with sex-specific deaths. A similar finding was also documented in India [31].

## Strengths and weakness

This study has several distinctive strengths. This study utilised cemetery-based death datasets to assess the excess mortality during pandemic, which are rarely used in the current research in Bangladesh. Second, this study utilised the most recent data from an urban region of Bangladesh (i.e., Jamalpur town) and produced findings that may be useful for future policy formulation and research. Finally, this study used a unique statistical approach that is more robust than typical time-series models. This study has also some limitations. First, it was not possible to determine the number of burials from this region that took place in cemeteries other than the study cemetery, which might lead to underreporting. Second, no covariates were evaluated due to a lack of data. Thirdly, even though BSTS models produced better results than conventional models, the inherent uncertainty of the data may impact the accuracy of

predictions. Finally, It should be noted that this dataset has not been validated for estimating excess mortality in this specific setting. Therefore, it is essential to exercise caution when interpreting the results and to consider the limitations and potential biases of the data. It is worth noting that the purpose of this study was not to provide a 100% accurate prediction and impact, but rather to present policymakers with vital information so that they may plan their strategies properly, and for researchers to use this type of data in future studies. Our findings cannot be generalised to every urban setting in the country. However, Jamalpur town's socio-demographic factors are comparable to those of a large number of district towns in Bangladesh [14]. This study does not provide information on COVID-19-related mortality; rather, it provides information regarding all-cause mortality during the COVID-19 period. Finding of this study, in our opinion, will aid LMICs like Bangladesh in effectively prioritising policies to improve death registration systems in order to better understand the mortality trend and put them into action.

## Conclusions

This study provides an essential comparison of mortality before and during the COVID-19 pandemic in order to understand the excess mortality and illustrates trends based on a cemetery-based dataset from an urban region of Bangladesh. Our findings indicate a pattern of increasing mortality during the pandemic; however, a high level of uncertainty surrounding the effect estimates derived from model instructs us to interpret effect estimates with caution. In addition, a lack of information prevented us from determining whether or not these deaths were directly or indirectly caused by the COVID-19 pandemic. It appears that cemetery-based death registration data might aid in the tracking of population mortality, particularly in resource-limited nations where collecting data on the ground is difficult during crisis moments. This dataset is a valuable resource for future research and policy development, but larger samples from multiple cemeteries are needed. This study's findings contribute to the scientific literature by expanding our understanding of how the policies of different countries have influenced the spread and severity of the pandemic.

## Supporting information

**S1 Table. The difference in the monthly average number of deaths between pre-COVID-19 and COVID periods, as well as a model-based estimate of the impact of COVID-19.**
(DOCX)

## Acknowledgments

We thank Dr. Pronoy Kanti Das, Civil Surgeon, Jamalpur General Hospital for support at the community level.

## Author Contributions

**Conceptualization:** Mohammad Sorowar Hossain, Enayetur Raheem.

**Data curation:** Jahidur Rahman Khan, S. M. Abdullah Al Mamun.

**Formal analysis:** Jahidur Rahman Khan.

**Investigation:** Mohammad Sorowar Hossain.

**Methodology:** Mohammad Sorowar Hossain, Jahidur Rahman Khan, S. M. Abdullah Al Mamun, Mohammad Tariqul Islam, Enayetur Raheem.

**Project administration:** Mohammad Sorowar Hossain.

**Resources:** Mohammad Sorowar Hossain, S. M. Abdullah Al Mamun, Mohammad Tariqul Islam.

**Supervision:** Mohammad Sorowar Hossain, Enayetur Raheem.

**Visualization:** Jahidur Rahman Khan, Mohammad Tariqul Islam.

**Writing – original draft:** Mohammad Sorowar Hossain, Jahidur Rahman Khan.

**Writing – review & editing:** Jahidur Rahman Khan, S. M. Abdullah Al Mamun, Mohammad Tariqul Islam, Enayetur Raheem.

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
