## [Decision Letter · Decision Letter 0]

13 Mar 2023

PGPH-D-22-01903

Excess mortality associated with the COVID-19 pandemic (2020-2021) in an urban community of Bangladesh

Dear Dr. Hossain,

Thank you for submitting your manuscript to PLOS Global Public Health. After careful consideration, we feel that it has merit but does not fully meet PLOS Global Public Health’s publication criteria as it currently stands. Therefore, we invite you to submit a revised version of the manuscript that addresses the points raised during the review process.

The manuscript has been evaluated by one reviewer, and their comments are available below. The reviewer has raised major concerns regarding the methodology, reporting and statistical analysis of this study. Could you please revise the manuscript to carefully address the concerns raised?

Please note that we have only been able to secure a single reviewer to assess your manuscript. We are issuing a decision on your manuscript at this point to prevent further delays in the evaluation of your manuscript. Please be aware that the editor who handles your revised manuscript might find it necessary to invite additional reviewers to assess this work once the revised manuscript is submitted. However, we will aim to proceed on the basis of this single review if possible.

We look forward to receiving your revised manuscript.

Kind regards,

Johannes Stortz

Staff Editor

Journal Requirements:

1. Please provide separate figure files in .tif or .eps format only and remove any figures embedded in your manuscript file. Please also ensure that all files are under our size limit of 10MB.

Additional Editor Comments (if provided):

Reviewers' comments:

Reviewer's Responses to Questions

**Comments to the Author**

1. Does this manuscript meet PLOS Global Public Health’s publication criteria? Is the manuscript technically sound, and do the data support the conclusions? The manuscript must describe methodologically and ethically rigorous research with conclusions that are appropriately drawn based on the data presented.

Reviewer #1: Partly

2. Has the statistical analysis been performed appropriately and rigorously?

Reviewer #1: I don't know

3. Have the authors made all data underlying the findings in their manuscript fully available (please refer to the Data Availability Statement at the start of the manuscript PDF file)?

Reviewer #1: No

4. Is the manuscript presented in an intelligible fashion and written in standard English?

Reviewer #1: Yes

5. Review Comments to the Author

Reviewer #1: This is an interesting and important article examining excess mortality during the COVID-19 pandemic in Bangladesh. Authors use both traditional comparisons as well as a Bayesian time series model to examine the all cause mortality in a single cemetery in Bangladesh. The study reports no significant change in morality during the pandemic and describes results in the context of the setting and data sources. A few points should be clarified to provide additional methodological information and highlight study limitations.

Major:

1. Lines 96-97, given this study is only examining cemetery burials in a specific town the assumption that individuals are buried in their local community—and that this practice did not change during the pandemic— has important implications for the results. I would request that the authors acknowledge this limitation more strongly in their discussion. I also request that authors highlight more strongly that these data have not been validated for use of excess mortality estimates in this setting.

2. The authors use a BSTS interventional time series model, which seems appropriate for the aim. However, additional details in the methods regarding the model selection—including the functional form used in the Bayesian model, what model predictors were included in the forecasting of the counterfactual, etc. I would ask authors to provide more detailed information on the modelling process and factors that were included in the model.

3. While the Wilcoxon rank-sum tests indicate a significant difference between the pre-pandemic and pandemic periods, these are not congruent with the regression analyses. In both the sex-specific and overall regression analyses the credible intervals cross 0 and are extremely wide. While the authors indicate that the regression results are not statistically significant, they assert in the abstract that COVID-19 did contribute to excess mortality. This is well described in the discussion, and I would ask authors to use similar language in their abstract.

4. Similarly, in the discussion I would ask authors to unpack this further and provide additional context, given that the model-based estimates indicate that there could have been an increase or decrease (based on the wide confidence intervals). Perhaps adding additional context in the limitation on this dataset, particularly since it has not be validated, would be helpful. The authors point about road traffic accidents decreasing and this offsetting COVID-19 related deaths, is important, and warrants further attention. Authors may also comment on the role of vaccines.

Minor:

The manuscript could benefit from an additional read through for grammatical and spelling errors and to provide clearer context, some of these instances are highlighted below.

1. Lines 54-55, please revise this sentence – perhaps stating “increase in all-cause morality during the COVID-19 pandemic” instead of using the word including.

2. Line 68, could authors provide some context on what they mean by “average urban setting”

3. Line 82 – I believe authors mean ‘census data’ not consensus data.

4. This is perhaps a minor point, but the authors switch between outbreak, epidemic, and pandemic throughout the manuscript. Please be consistent in how these are used.

Additional methodological minor comments are included below

5. Lines 104-105, it is unclear why authors have excluded those under 35. This seems like it would be an important demographic to see differences, in particular since authors note that official records were not reliable during this period. I would request authors provide further context on why they excluded this, or to conduct a sensitivity analysis by including these individuals.

6. In figure 1 please update the flow chart to show the exclusion of deaths that were outside of the study time frame.

6. PLOS authors have the option to publish the peer review history of their article (what does this mean?). If published, this will include your full peer review and any attached files.

**Do you want your identity to be public for this peer review?** For information about this choice, including consent withdrawal, please see our Privacy Policy.

Reviewer #1: No

---

## [Editor Report · Decision Letter 1]

22 Jun 2023

Excess mortality during the COVID-19 pandemic (2020-2021) in an urban community of Bangladesh

PGPH-D-22-01903R1

Dear Dr Hossain,

We are pleased to inform you that your manuscript 'Excess mortality during the COVID-19 pandemic (2020-2021) in an urban community of Bangladesh' has been provisionally accepted for publication in PLOS Global Public Health.

Best regards,

Ting Shi

Academic Editor